# CROSS-MODAL FACTOR REASONING WITH LLMS: TOWARD SEMANTIC-STRUCTURED GENERALIZATION FOR RECOMMENDATION

## ABSTRACT

Multimodal recommendation aims to enhance personalization by leveraging content signals such as text and images. However, existing methods often treat modalities as shallow auxiliary inputs, fusing raw embeddings without reasoning about what semantics are useful or how they influence user preference. Content-based graphs typically rely on low-level similarity, lacking structured semantic relations such as functionality or style. Moreover, collaborative signals are used solely for ranking, without grounding content semantics. To address these limitations, we present MARS, a framework for Cross-Modal FActor Reasoning with LLMs, enabling Semantic-Structured Generalization in recommendation. MARS introduces a cognitively guided paradigm that prompts large language models (LLMs) to extract human-interpretable semantic factors (e.g., functionality, material and usage scenario) from raw visual and textual descriptions. These structured factors are used to build heterogeneous graphs that capture multi-aspect semantic relations among items. To integrate semantics into representation learning, we propose an auxiliary semantic prediction task that aligns collaborative embeddings with LLM-inferred factor knowledge. In addition, a cross-modal consistency loss encourages agreement across semantic views from different modalities. Extensive experiments show that MARS achieves superior accuracy and generalization compared to state-of-the-art multimodal baselines and LLM-based methods.

## 1 INTRODUCTION

Multimodal recommendation Liu et al. (2024d); Malitesta et al. (2024); Liu et al. (2024b) has emerged as a powerful paradigm for enhancing personalized recommendation by incorporating heterogeneous content signals such as textual descriptions and visual appearances into the learning process. In addition to mitigating cold-start issues, multimodal inputs provide rich semantic context that helps uncover user preferences and item characteristics Han et al. (2022); Zhou et al. (2023a). This makes multimodal recommendation especially valuable in content-rich domains like e-commerce, social media and multimedia platforms where user decisions are shaped by the interaction of diverse modalities. As a result, increasing research has focused on leveraging multimodal information to improve representation learning Cen et al. (2020); He et al. (2021).

Despite substantial progress, current approaches to multimodal recommendation remain limited in their ability to model content semantics. Most methods encode item modalities such as images and text using pretrained models, then inject these features into collaborative filtering frameworks through concatenation He & McAuley (2016), gating Xiao et al. (2022), attention Ji et al. (2023), or graph-based propagation Tao et al. (2020); Liu et al. (2023); Guo et al. (2024). These approaches typically treat multimodal features as auxiliary signals, using them as generic vectors without considering their underlying semantic meaning Zhang et al. (2021); Jiang et al. (2024); Ong & Khong (2024). As a result, the learning process is often shallow, combining low-level representations without interpreting the specific semantic roles that each modality plays in shaping user preferences. This leads to a key unresolved question: *which* semantic cues from each modality are truly useful for recommendation, and *how* can models selectively reason over them?

These limitations become more pronounced when modality-based item-item relations are constructed for graph modeling. Although prior works have explored visual or textual similarity to build item-item graphs, these relations are typically based on low-level embedding proximity Liu et al. (2024c); Zhou & Shen (2023). Such designs fail to capture structured semantics such as item complementarity, stylistic coherence or functional substitutability, which users often perceive implicitly. Moreover, they tend to overlook the task-specific relevance of different semantic dimensions. For example, two items may appear visually similar yet serve entirely unrelated purposes in a given context. The inability to encode such high-level semantic structure limits the expressiveness of multimodal graphs and weakens their ability to support fine-grained preference reasoning.

An even deeper issue lies in the one-way flow of supervision from user behavior to content modeling. Most existing frameworks use multimodal content merely as auxiliary input for collaborative ranking, while the collaborative signal itself encoded in rich user-item interaction data is seldom utilized to refine or supervise content understanding Wu et al. (2024). In other words, content supports recommendation but recommendation rarely informs content interpretation. This creates a fundamental semantic gap where model representations may succeed in ranking tasks yet remain opaque and semantically ungrounded. Addressing this gap requires rethinking the role of collaborative signals not only as behavioral evidence but also as supervision for semantic representation learning.

To address the above challenges, we propose MARS, a novel framework for Cross-Modal Factor Reasoning with LLMs that enables semantic-aware graph learning for recommendation. Instead of shallow encoding or early fusion, MARS adopts a cognitively informed paradigm that leverages large language models (LLMs) to extract high-level, human-understandable semantic factors (e.g., functionality, material) from raw visual and textual content. These interpretable factors are used to assign each item a structured attribute profile, which forms the basis for three heterogeneous graphs, respectively constructed from textual, visual, and ID-derived semantics. We then perform factor-wise graph propagation via lightweight GCNs and introduce a dynamic fusion mechanism that allows each item to adaptively integrate information across graphs based on its content. To enhance bidirectional semantic alignment, MARS introduces an auxiliary semantic prediction task that aligns collaborative embeddings with LLM-inferred semantics. In addition, a cross-view consistency regularization is applied to ensure coherence across modality-specific representations.

The main contributions can be summarized as follows:

- We propose a novel LLM-driven multimodal recommendation framework that leverages LLMs to perform high-level semantic factor reasoning. This enables the construction of structured heterogeneous graphs that capture fine-grained semantic relations among items.

- We introduce a semantic factor prediction auxiliary task that supervises collaborative representations using LLM-inferred semantic knowledge. This empowers the learned embeddings with human-understandable semantics.

- Comprehensive evaluations on Amazon benchmarks demonstrate that our method consistently outperforms competitive baselines across various settings.

Moreover, we provide a more detailed discussion of related work in Appendix A.

## 2 THE PROPOSED MODEL

Fig. 1 illustrates the overall architecture of our proposed framework, which centers around LLM-driven semantic factor reasoning, heterogeneous graph construction, and semantically grounded representation learning.

### 2.1 LLM-DRIVEN CROSS-MODAL FACTOR REASONING AND HETEROGENEOUS GRAPH CONSTRUCTION

Existing multimodal recommendation methods often rely on pretrained encoders or self-supervised alignment, representing visual and textual content as dense but semantically opaque vectors. This lack of explicit semantic structure limits interpretability and weakens the alignment between content features and user preferences. Therefore, we introduce a framework that leverages large language

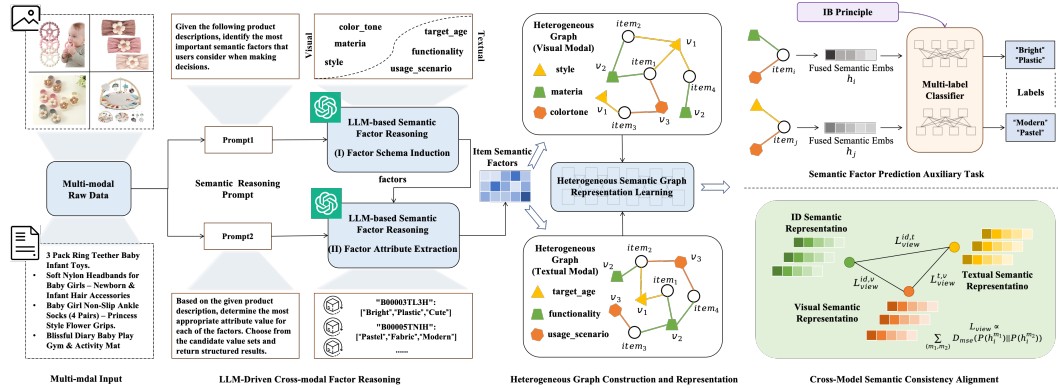

Figure 1: The overall architecture of our proposed framework. It integrates LLM-driven semantic factor reasoning, heterogeneous graph construction, and multi-view collaborative modeling to enhance multimodal recommendation.

models (LLMs) to extract structured, high-level semantic factors from multimodal inputs. Specifically, LLMs are prompted to identify (1) the key semantic dimensions relevant to recommendation intent (e.g., Functionality, Style, Material), and (2) the corresponding attribute values for each item under these dimensions (e.g., Functionality = Outdoor, Style = Cartoon). Based on this information, we construct heterogeneous item graphs for each modality, where edges capture factor-level semantic relations. These graphs move beyond simple similarity-based connections and encode human-interpretable semantics, enhancing both the expressiveness and transparency.

### 2.1.1 LLM-BASED SEMANTIC FACTOR REASONING

In light of the above analysis, we propose a novel perspective: leveraging the cognitive and symbolic reasoning capabilities of Large Language Models (LLMs) to construct a factor-based semantic representation space. This empowers the model to move from passive feature consumption to active content understanding. Specifically, we design a two-stage LLM-driven reasoning framework to extract structured semantic factor representations from item-level multimodal content. Consider the case of text input,

**Stage 1: Factor Schema Induction.** We begin by prompting the LLM to identify a set of key semantic factors that govern user decision-making in the target domain:

> *"Given the following product descriptions, identify the most important semantic factors*
> *that users consider when making decisions. Each factor should capture a different perspec-*
> *tive, such as functionality, material, or usage scenario."*

The LLM returns a set of interpretable and domain-relevant factor names:

$$\mathcal{F} = \{f_1, f_2, \ldots, f_K\}. \tag{1}$$

Each factor $f_k$ is associated with a finite attribute space $\mathcal{A}_{f_k}$, forming a structured semantic schema.

**Stage 2: Factor Attribute Assignment.** For each item $i \in \mathcal{I}$, we query the LLM to infer its attribute values under the identified schema $\mathcal{F}$. The multimodal content of each item $\mathbf{Z}_i = \{\mathbf{z}_i^v, \mathbf{z}_i^t\}$ is transformed into textual descriptions (e.g. captions) and fed into the LLM using the instruction:

> *"Based on the given product description, determine the most appropriate attribute value for*
> *each of the following factors: [Functionality, Style, Material]. Choose from the candidate*
> *value sets and return structured results."*

The output is a structured profile:

$$\mathcal{A}_i = \{(f_k, a_k^{(i)}) \mid f_k \in \mathcal{F}, a_k^{(i)} \in \mathcal{A}_{f_k}\}. \tag{2}$$

This LLM-based factor reasoning paradigm enables us to construct a semantic graph space that is interpretable, extensible, and inherently aligned with human cognitive priors.

### 2.1.2 HETEROGENEOUS SEMANTIC GRAPH CONSTRUCTION

Based on the semantic factor profiles $\mathcal{A}_i = \{(f_k, a_k^{(i)})\}$ inferred by LLMs, we construct a set of heterogeneous item-item graphs that capture the semantic proximity between items from multiple complementary perspectives. Unlike conventional item graphs built from visual or textual embeddings using similarity metrics (e.g., cosine or Euclidean distance), our graphs are structured upon interpretable symbolic attributes, enabling fine-grained semantic relation modeling.

**Factor-wise Graphs.** For each semantic factor $f_k \in \mathcal{F}$, we define a graph $\mathcal{G}_{f_k} = (\mathcal{I}, \mathcal{E}_{f_k})$, where nodes represent items and an edge $(i, j) \in \mathcal{E}_{f_k}$ exists if and only if items $i$ and $j$ share the same attribute value under factor $f_k$:

$$(i, j) \in \mathcal{E}_{f_k} \Longleftrightarrow a_k^{(i)} = a_k^{(j)}. \tag{3}$$

This induces a factor-specific semantic clustering of items, where edges represent human-understandable equivalence under certain cognitive dimensions (e.g., "both are waterproof", etc.). Each $\mathcal{G}_{f_k}$ forms a homogeneous subgraph under the semantic lens of $f_k$, while the collection $\{\mathcal{G}_{f_k}\}_{k=1}^K$ together constitute a heterogeneous semantic graph system.

**Adjacency Matrix Encoding.** Each graph is encoded by a binary adjacency matrix $\mathbf{A}_{f_k} \in \{0, 1\}^{|\mathcal{I}| \times |\mathcal{I}|}$, where $\mathbf{A}_{f_k}(i, j) = 1$ if $a_k^{(i)} = a_k^{(j)}$, and 0 otherwise. To enable graph propagation, we apply symmetric normalization:

$$\hat{\mathbf{A}}_{f_k} = \mathbf{D}_{f_k}^{-\frac{1}{2}} \mathbf{A}_{f_k} \mathbf{D}_{f_k}^{-\frac{1}{2}}, \tag{4}$$

where $\mathbf{D}_{f_k}$ denotes the degree matrix of $\mathbf{A}_{f_k}$. This approach transitions from superficial content similarity toward structured semantic understanding.

### 2.1.3 MULTI-GRAPH ENHANCED ITEM REPRESENTATION LEARNING

Given the heterogeneous semantic graphs $\{\hat{\mathbf{A}}_{f_k}\}_{k=1}^K$, we enhance item representations through multi-graph propagation. In addition to a structural graph (e.g., user-item or content similarity), our model incorporates semantic graphs to capture complementary cognitive signals.

**Backbone Graph Encoding.** We begin with a backbone graph encoder that operates on the unified multi-modal representation of items. Specifically, we construct the base item representation $\mathbf{X} = [\mathbf{X}_1; \mathbf{X}_2; ...; \mathbf{X}_{|L|}] \in \mathbb{R}^{|\mathcal{I}| \times d}$ by integrating the modality-specific features:

$$\mathbf{X}_i = \text{Concat}\left(\mathbf{z}_i^v, \mathbf{z}_i^t, \mathbf{z}_i^{\text{id}}\right) \cdot \mathbf{W}_0, \tag{5}$$

where $\mathbf{W}_0 \in \mathbb{R}^{(d_v + d_t + d_{id}) \times d}$ projects concatenated features into a unified space. On top of $\mathbf{X}$, we apply standard graph convolution over the graph $\hat{\mathbf{A}}_{\text{struct}}$ to obtain structural-enhanced embeddings: $\mathbf{H}_{\text{struct}} = \hat{\mathbf{A}}_{\text{struct}} \mathbf{X}$. Here, $\hat{\mathbf{A}}_{\text{struct}}$ is derived from the fully connected graph of item-wise similarities by retaining only the top-K edges, which is consistent with previous works Meng et al. (2025); Dang et al. (2025).

**Semantic Graph Propagation.** For each semantic factor $f_k$, we independently propagate over its semantic graph:

$$\mathbf{H}_{f_k} = \hat{\mathbf{A}}_{f_k} \mathbf{X}. \tag{6}$$

This results in $K$ semantic-aware embeddings $\{\mathbf{H}_{f_k}\}_{k=1}^K$, each capturing item relationships from a specific semantic view. These representations are conceptually orthogonal and encode disentangled semantics.

**Adaptive Fusion of Multi-Graph Signals.** To effectively integrate heterogeneous semantics into item embeddings, we propose an item-level adaptive fusion mechanism:

$$\mathbf{H}_i = \sum_{k=1}^K \alpha_i^{(k)} \cdot \mathbf{H}_{f_k}(i) + \alpha_i^{(\text{struct})} \cdot \mathbf{H}_{\text{struct}}(i), \tag{7}$$

where $\alpha_i^{(k)}$ denotes the importance weight of semantic factor $f_k$ for item $i$, and $\alpha_i^{(\text{struct})}$ for structural signal. These coefficients are learnable and normalized via softmax:

$$[\alpha_i^{(\text{struct})}, \alpha_i^{(1)}, ..., \alpha_i^{(K)}] = \text{softmax}(\mathbf{W}_g \cdot \mathbf{X}_i), \tag{8}$$

where $\mathbf{W}_g \in \mathbb{R}^{(K+1)\times d}$ is a gating matrix. This design enables each item to adaptively prioritize structural vs. semantic signals, depending on its own content and graph context. Please refer to Appendix A for model complexity analysis.

## 2.2 SEMANTIC FACTOR PREDICTION AUXILIARY TASK: ENRICHING COLLABORATIVE SIGNALS WITH SEMANTICS

While the heterogeneous graphs provide a structured prior, the semantic prediction task is the critical mechanism that ensures the collaborative embeddings are semantically grounded. It forces the model to learn representations that are not only effective for ranking but are also decodable into the human-understandable factor space established by the LLM. This explicitly bridges the gap between sub-symbolic collaborative patterns and symbolic semantic knowledge. In our framework, we propose a novel auxiliary task that explicitly aligns collaborative embeddings with high-level semantic attributes, effectively endowing them with interpretable and cognitively meaningful capabilities.

**Problem Formulation.** Let $\mathcal{F} = \{f_1, f_2, \ldots, f_K\}$ denote the set of semantic factors derived from LLM-guided reasoning (e.g., functionality, material, usage scenario). Each item $i$ is associated with a multi-hot label vector $\mathbf{y}_i \in \{0, 1\}^{|\mathcal{A}|}$, where $\mathcal{A} = \bigcup_k \mathcal{A}_{f_k}$ is the union of all attribute values across semantic factors. For example, if $f_1$ is "Material", then $\mathcal{A}_{f_1} = \{\text{cotton, wood, plastic, ...}\}$.

Our goal is to learn a semantic decoder $\psi : \mathbb{R}^d \to [0, 1]^{|\mathcal{A}|}$ that maps item embeddings into the semantic attribute space. We implement $\psi(\cdot)$ as a lightweight multi-label classifier (e.g., linear projection or MLP), operating on the final multi-graph enhanced item embedding $\mathbf{H}_i$. The prediction is formulated as:

$$\hat{\mathbf{y}}_i = \mathrm{sigmoid}(\psi(\mathbf{H}_i)). \tag{9}$$

**Auxiliary Training Objective.** We adopt a binary cross-entropy loss to train the semantic prediction head:

$$\mathcal{L}_{\mathrm{sem}} = -\frac{1}{|\mathcal{I}|} \sum_{i \in \mathcal{I}} \sum_{j=1}^{|\mathcal{A}|} \left[ y_{ij} \log \hat{y}_{ij} + (1 - y_{ij}) \log(1 - \hat{y}_{ij}) \right]. \tag{10}$$

**Theoretical Insight: An Information-Theoretic View.** The proposed auxiliary task can be interpreted as a semantic grounding mechanism that transforms item representations from behavioral memory spaces into semantic spaces:

$$\mathrm{Embedding}_{\mathrm{collab}} \xrightarrow{\text{Semantic Prediction}} \mathrm{Embedding}_{\mathrm{semantic}}$$

Specifically, we aim to increase the mutual information between the learned embedding $\mathbf{H}_i$ and the semantic attribute vector $\mathbf{y}_i$, while compressing unnecessary noise. This aligns with the information bottleneck principle Tishby et al. (2000), formulated as:

$$\max_{\psi} \ I(\mathbf{H}_i; \mathbf{y}_i) - \beta I(\mathbf{H}_i; \mathbf{z}_i^{\mathrm{raw}}), \tag{11}$$

where $\mathbf{z}_i^{\mathrm{raw}}$ denotes the high-dimensional raw inputs (e.g., vision/text features), and $\beta$ is a trade-off factor. The optimization encourages the embedding $\mathbf{H}_i$ to retain semantic-relevant information while filtering out modality-specific noise. In practice, our multi-label classification loss $\mathcal{L}_{\mathrm{sem}}$ serves as a lower bound surrogate to maximize $I(\mathbf{H}_i; \mathbf{y}_i)$, enforcing that collaborative signal not only encodes interaction patterns but also preserves semantic discriminability. Please refer to Appendix A for theoretical analysis.

## 2.3 CROSS-MODAL VIEW CONSISTENCY REGULARIZATION

While ID, visual, and textual features offer complementary views of items, their heterogeneous distributions often lead to misaligned representations. For instance, visual embeddings may cluster by color or texture, textual embeddings by syntactic semantics, and ID embeddings by collaborative signals. Such discrepancies can cause training instability and degrade generalization. To mitigate this, we introduce a lightweight cross-view consistency loss to align modalities.

### 2.3.1 CROSS-MODAL VIEW CONSISTENCY LOSS

Let $\mathbf{H}_i^{\text{id}}, \mathbf{H}_i^{\text{v}}, \mathbf{H}_i^{\text{t}}$ denote the ID, visual, and textual embeddings of item $i$, respectively. Although each captures distinct information, we encourage their representations to converge semantically. We define the cross-view consistency loss as:

$$\mathcal{L}_{\text{view}} = \frac{1}{|\mathcal{I}|} \sum_{i \in \mathcal{I}} \left[ \|\mathbf{H}_i^{\text{id}} - \mathbf{H}_i^{\text{v}}\|^2 + \|\mathbf{H}_i^{\text{id}} - \mathbf{H}_i^{\text{t}}\|^2 + \|\mathbf{H}_i^{\text{v}} - \mathbf{H}_i^{\text{t}}\|^2 \right]. \tag{12}$$

This regularization encourages modality-invariant embeddings and stabilizes training across views.

### 2.4 SEMANTIC-ENHANCED MULTI-OBJECTIVE LEARNING

Our model aims to learn a scoring function $s(f_u(u), g_i(i))$ that ranks positive user-item interactions above negatives, where $f_u(u)$ and $g_i(i)$ denote the user / item ID embedding. We adopt a pairwise learning framework with the Bayesian Personalized Ranking (BPR) loss as the main objective $\mathcal{L}_{\text{rec}}$.

To jointly promote semantic discrimination and cross-modal alignment, we incorporate two auxiliary objectives: semantic prediction loss $\mathcal{L}_{\text{sem}}$ and view consistency loss $\mathcal{L}_{\text{view}}$. The final optimization target becomes:

$$\mathcal{L} = \mathcal{L}_{\text{rec}} + \lambda \cdot \mathcal{L}_{\text{sem}} + \eta \cdot \mathcal{L}_{\text{view}}, \tag{13}$$

where $\lambda$ and $\eta$ are hyperparameters balancing the influence of auxiliary tasks.

## 3 EXPERIMENTS

In this section, we present comprehensive experiments to assess the effectiveness and robustness of our framework, focusing on the following research questions: (**RQ1**) How does our model compare with state-of-the-art multimodal methods? (**RQ2**) Can it handle cold-start cases while preserving recommendation quality? (**RQ3**) How do key components affect overall performance? (**RQ4**) How sensitive is the model to hyperparameters such as semantic weight and embedding size? (**RQ5**) Does it reduce inter-modal divergence and improve semantic consistency? (**RQ6**) Can it uncover factor-aware semantic structures under explicit supervision?

### 3.1 EXPERIMENTAL SETTINGS

#### 3.1.1 DATASETS

We evaluate our framework on three standard subsets of the Amazon Review dataset—*Baby*, *Sports and Outdoors*, and *Clothing, Shoes and Jewelry*. Rather than using handcrafted or pretrained features, we feed these raw multimodal inputs into LLMs to perform structured semantic reasoning. Please refer to Appendix B for more details.

#### 3.1.2 BASELINES

We compared MARS with the following baselines. Collaborative filtering methods: **LightGCN** He et al. (2020). Multimodal methods: **MMGCN** Wei et al. (2019), **GRCN** Wei et al. (2020), **DualGNN** Wang et al. (2021), **LATTICE** Zhang et al. (2021), **BM3** Zhou et al. (2023b), **FREEDOM** Zhou & Shen (2023), **DiffMM** Jiang et al. (2024), **MMIL** Yang & Yang (2024), **AlignRec** Liu et al. (2024c), **SMORE** Ong & Khong (2025). LLM-based baselines: **RecFormer** Li et al. (2023), **TALLRec** Bao et al. (2023), **A-LLMRec** Kim et al. (2024) and **UniMP** Wei et al. (2024a).

#### 3.1.3 IMPLEMENTATION DETAILS

We implement our framework using PyTorch and build upon the MMRec Zhou (2023) codebase to ensure consistent and reproducible comparisons with prior multimodal recommendation methods. The textual and visual embedding are initialized with Sentence-BERT and LLaVA-7B. For semantic factor reasoning, we employ the GPT-4o model as the backbone LLM to extract structured semantics from raw inputs. All experiments are conducted on a single NVIDIA A40 GPU (48GB). We use Recall@$K$ and NDCG@$K$ for evaluation. Please refer to Appendix B for all settings.

Table 1: Performance comparison on three Amazon domains. R and N denote Recall and NDCG, respectively. MARS achieves consistently superior results over competitive multimodal baselines across all categories.

| Dataset | Metric | Traditional | Classical Multimodal | | | | Advanced Multimodal | | | | Large Language Models-based | | | | Ours |
|---|---|---|---|---|---|---|---|---|---|---|---|---|---|---|---|
| | | LightGCN | MMGCN | DualGNN | LATTICE | FREEDOM | DiffMM | MMIL | AlignRec | SMORE | RecFormer | TALLRec | A-LLMRec | UniMP | MARS |
| Sports | R@10 | 0.0569 | 0.0394 | 0.0574 | 0.0628 | 0.0705 | 0.0687 | 0.0747 | 0.0758 | 0.0762 | 0.0375 | 0.0418 | 0.0402 | 0.0528 | 0.0805 |
| | R@20 | 0.0864 | 0.0625 | 0.0881 | 0.0961 | 0.1077 | 0.1035 | 0.1133 | 0.1160 | 0.1142 | 0.0608 | 0.0674 | 0.0703 | 0.0757 | 0.1198 |
| | N@10 | 0.0311 | 0.0203 | 0.0316 | 0.0339 | 0.0382 | 0.0357 | 0.0405 | 0.0414 | 0.0408 | 0.0178 | 0.0247 | 0.0223 | 0.0288 | 0.0436 |
| | N@20 | 0.0387 | 0.0266 | 0.0393 | 0.0431 | 0.0478 | 0.0458 | 0.0505 | 0.0517 | 0.0506 | 0.0237 | 0.0298 | 0.0374 | 0.0399 | 0.0541 |
| Clothing | R@10 | 0.0361 | 0.0221 | 0.0447 | 0.0503 | 0.0616 | 0.0593 | 0.0643 | 0.0651 | 0.0659 | 0.0384 | 0.0397 | 0.0458 | 0.0463 | 0.0702 |
| | R@20 | 0.0544 | 0.0357 | 0.0663 | 0.0755 | 0.0917 | 0.0874 | 0.0961 | 0.0993 | 0.0987 | 0.0645 | 0.0673 | 0.0682 | 0.0721 | 0.1025 |
| | N@10 | 0.0197 | 0.0116 | 0.0237 | 0.0277 | 0.0333 | 0.0325 | 0.0348 | 0.0356 | 0.0360 | 0.0178 | 0.0188 | 0.0192 | 0.0256 | 0.0382 |
| | N@20 | 0.0243 | 0.0151 | 0.0289 | 0.0356 | 0.0409 | 0.0396 | 0.0428 | 0.0437 | 0.0443 | 0.0243 | 0.0267 | 0.0283 | 0.0324 | 0.0463 |
| Baby | R@10 | 0.0479 | 0.0413 | 0.0507 | 0.0561 | 0.0624 | 0.0617 | 0.0670 | 0.0674 | 0.0680 | 0.0348 | 0.0382 | 0.0379 | 0.0472 | 0.0709 |
| | R@20 | 0.0754 | 0.0649 | 0.0782 | 0.0867 | 0.0985 | 0.0978 | 0.1035 | 0.1046 | 0.1035 | 0.0535 | 0.0687 | 0.0597 | 0.0718 | 0.1089 |
| | N@10 | 0.0257 | 0.0211 | 0.0264 | 0.0305 | 0.0324 | 0.0321 | 0.0361 | 0.0363 | 0.0365 | 0.0172 | 0.0197 | 0.0203 | 0.0267 | 0.0380 |
| | N@20 | 0.0328 | 0.0275 | 0.0335 | 0.0383 | 0.0416 | 0.0408 | 0.0455 | 0.0458 | 0.0457 | 0.0219 | 0.0256 | 0.0238 | 0.0337 | 0.0483 |

Table 2: Cold-start results for users with 5 interactions in three Amazon categories. For brevity, only a subset of representative baselines are shown. MARS consistently delivers top-tier performance across all evaluated settings and metrics.

| Model | Amazon-Sports | | | | Amazon-Clothing | | | | Amazon-Baby | | | |
|---|---|---|---|---|---|---|---|---|---|---|---|---|
| | R@10 | R@20 | N@10 | N@20 | R@10 | R@20 | N@10 | N@20 | R@10 | R@20 | N@10 | N@20 |
| GRCN | 0.0556 | 0.0814 | 0.0305 | 0.0370 | 0.0416 | 0.0648 | 0.0213 | 0.0271 | 0.0510 | 0.0770 | 0.0268 | 0.0333 |
| BM3 | 0.0581 | 0.0920 | 0.0305 | 0.0390 | 0.0424 | 0.0614 | 0.0229 | 0.0276 | 0.0549 | 0.0845 | 0.0288 | 0.0363 |
| LATTICE | 0.0380 | 0.0570 | 0.0205 | 0.0252 | 0.0414 | 0.0555 | 0.0218 | 0.0254 | 0.0570 | 0.0827 | 0.0298 | 0.0362 |
| FREEDOM | 0.0622 | 0.0933 | 0.0328 | 0.0406 | 0.0444 | 0.0671 | 0.0241 | 0.0298 | 0.0535 | 0.0880 | 0.0297 | 0.0384 |
| MMIL | 0.0759 | 0.1141 | 0.0422 | 0.0517 | 0.0635 | 0.0948 | 0.0341 | 0.0423 | 0.0672 | 0.1014 | 0.0377 | 0.0463 |
| SMORE | 0.0788 | 0.1158 | 0.0420 | 0.0512 | 0.0676 | 0.0983 | 0.0369 | 0.0446 | 0.0687 | 0.1019 | 0.0381 | 0.0464 |
| **MARS** | **0.0841** | **0.1238** | **0.0445** | **0.0539** | **0.0703** | **0.1022** | **0.0379** | **0.0457** | **0.0740** | **0.1105** | **0.0397** | **0.0490** |

## 3.2 Overall Performance (RQ1)

As shown in Table 1, MARS consistently delivers state-of-the-art performance across all benchmarks and metrics. Compared to strong multimodal baselines such as SMORE and AlignRec, which rely on implicit fusion or hierarchical alignment, MARS achieves significant improvements by explicitly reasoning over high-level semantic factors and structurally incorporating them into graph-based learning. For instance, on the *Sports* dataset, MARS surpasses SMORE by 5.6% in Recall@10 and 5.0% in Recall@20, illustrating the effectiveness of structured semantics in enhancing item representations. Similar trends hold for *Clothing* and *Baby*, where properties like material and functionality play key roles in shaping user preferences. We further compare MARS with recent LLM-powered recommenders. RecFormer and TALLRec, though equipped with strong language priors, fall short due to their limited capacity to capture collaborative interaction patterns. A-LLMRec benefits from incorporating pretrained collaborative knowledge, yielding better alignment with user behavior. UniMP, a vision-language model designed for personalized tasks, shows improved cross-modal integration but still underperforms MARS. These comparisons suggest that without explicit semantic structuring and targeted supervision, general-purpose LLMs may struggle to effectively model recommendation-specific nuances.

## 3.3 Cold-Start Recommendation (RQ2)

We evaluate model performance in the user cold-start scenario, where each test user has exactly five historical interactions. As shown in Table 2, MARS consistently achieves the best results across all datasets and metrics, significantly surpassing all baselines. This validates our core intuition: although semantic factor reasoning operates at the item level, the structured semantics captured from multimodal content enable more informative representations that generalize well to sparse user histories. By propagating over high-level factor-aware graphs, MARS effectively aligns unseen users with semantically related items, even in the absence of dense interactions. Compared to methods like MMIL and SMORE, which exhibit some robustness through global interest modeling and frequency-domain fusion, MARS achieves stronger gains by injecting explicit semantic supervision into both the graph structure and training objectives. In contrast, methods such as FREEDOM experience larger performance drops under sparsity due to their reliance on shallow enhancement or dense interaction graphs. These results highlight the importance of structured semantic reasoning in addressing the cold-start challenge.

## 3.4 ABLATION STUDY (RQ3)

We conduct ablation studies to assess the impact of each core component in MARS by selectively removing or modifying individual modules (Fig. 2). Removing the semantic factor prediction task (`w/o FP`) or the cross-modal consistency loss (`w/o CL`) leads to clear performance drops, highlighting their roles in enforcing semantic supervision and modality alignment. Notably, the degradation from `w/o FP` confirms that semantic factors are not merely auxiliary inputs but serve as crucial supervisory signals during training. The removal of multimodal content (`w/o MM`) further degrades performance, emphasizing the necessity of visual and textual cues for capturing fine-grained semantics under sparse interactions. To examine semantic graph modeling, we evaluate three structural variants. Replacing heterogeneous semantic graphs with naive multimodal concatenation (`r/p SG`) results in substantial performance loss, indicating that shallow fusion lacks the capacity for semantic abstraction. More severely, replacing LLM-derived factors with unsupervised clustering (`r/p SF`) degrades performance even with graph and supervision signals, underscoring the importance of high-quality, LLM-guided semantics. Removing the semantic graph while maintaining prediction supervision (`w/o SG`) also weakens performance, showing that structural encoding of semantics, not just label supervision, is essential for effective knowledge transfer.

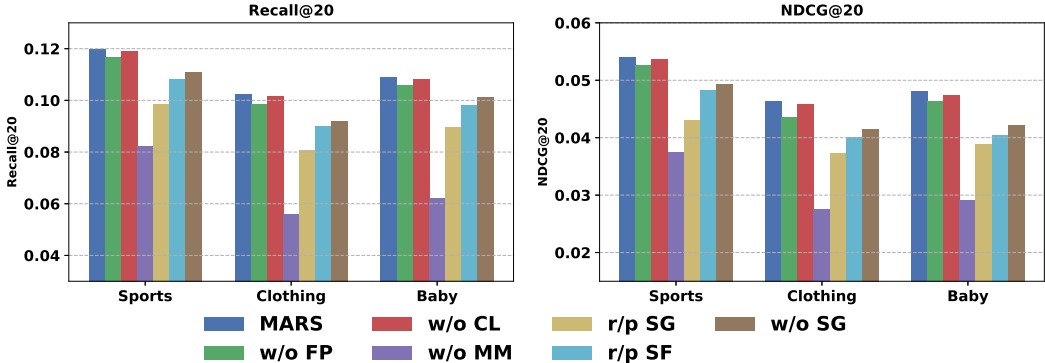

Figure 2: Results of the ablation study, illustrating the effectiveness of each individual component in MARS.

## 3.5 HYPERPARAMETER SENSITIVITY (RQ4)

To evaluate the robustness of our framework, we perform a sensitivity analysis on three critical hyperparameters: the weight $\lambda$ of the semantic prediction loss, the weight $\eta$ of the cross-modal view consistency loss, and the embedding dimension $d$. As shown in Fig. 3 and Fig. 4, we observe consistent trends across datasets. For $\lambda$, we find that strong supervision ($\lambda = 1$) degrades performance, likely due to interference with the collaborative objective. Performance improves as $\lambda$ decreases, peaking around $\lambda = 0.001$, indicating that lightweight semantic guidance effectively enhances interpretability without compromising utility. Similarly, a large consistency weight $\eta$ overly penalizes cross-view variations, whereas a moderate setting (e.g., $\eta = 0.01$) balances coherence and diversity across modalities, validating the effectiveness of our regularization design. We also analyze the impact of the embedding dimension $d$, as shown in Fig. 4 (right). Increasing $d$ from 32 to 128 yields steady performance gains, highlighting the benefit of higher representational capacity in capturing multimodal semantics. However, beyond $d = 128$, improvements diminish and may slightly fluctuate, suggesting potential overfitting or noise accumulation.

## 3.6 MEASURING INTER-MODAL REPRESENTATION DIVERGENCE (RQ5)

To further examine how our method enhances the alignment of multimodal semantics, we investigate the distribution of inter-view representation differences across items. Specifically, we calculate for each item the average pairwise L2 distance among the structural and multimodal representations, then compare the distributions of these divergence scores under two settings: with and without the proposed view consistency loss. As shown in Fig. 5, the experimental group that incorporates view-

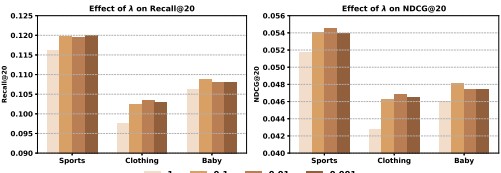
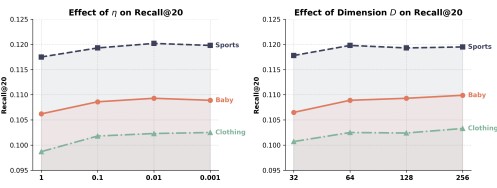

Figure 3: Effect of the semantic prediction loss weight $\lambda$ on recommendation performance across three datasets.

Figure 4: Impact of the cross-view consistency weight $\eta$ (left) and embedding dimension $d$ (right) on performance.

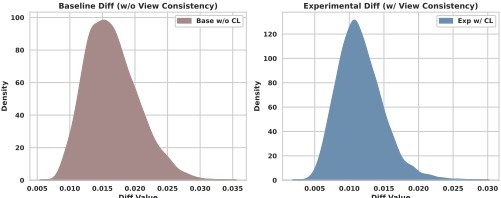
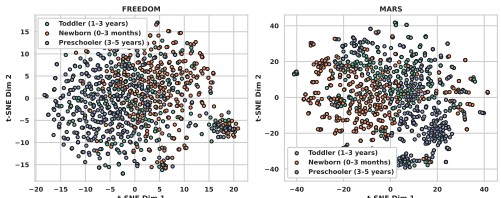

Figure 5: Distribution of representation divergence $Diff$ across views. Our approach yields significantly lower divergence, indicating more aligned multi-view representations.

Figure 6: t-SNE visualization of item embeddings colored by Target Age attribute. Our method (right) exhibits clearer semantic clustering compared to the baseline (left).

level regularization exhibits a significantly sharper and left-shifted distribution, suggesting more coherent and aligned item representations across modalities. This observation reveals that the proposed consistency constraint not only narrows the geometric gap between modality-specific views but also serves as a semantic calibrator that unifies how different modalities describe the same item. In effect, the model learns to emphasize modality-invariant semantics while suppressing noisy or conflicting signals. Such calibration promotes representation stability and robustness of preference modeling. These results validate the necessity of cross-view constraints and highlight the importance of consistent semantic grounding.

### 3.7 SEMANTIC STRUCTURE ANALYSIS UNDER FACTOR SUPERVISION (RQ6)

We evaluate the semantic alignment of learned embeddings through a factor-centric visualization on the *Target-Age* attribute. Specifically, we select three representative values, *Newborn (0–3 months)*, *Toddler (1–3 years)*, and *Preschooler (3–5 years)*, and color-code item embeddings using t-SNE. Fig. 6 compares the distributions from FREEDOM and our MARS. FREEDOM produces mixed clusters with no clear separation among semantic groups, indicating a lack of explicit factor modeling. In contrast, MARS (right) exhibits well-separated, compact clusters, especially for *Preschooler* items, demonstrating its ability to encode factor-aware structure. *While this visualization is qualitative, the improved structural coherence across semantic groups strongly implies successful semantic grounding.* This reflects the impact of our auxiliary semantic prediction task, which enforces alignment between collaborative signals and LLM-inferred semantics. These results support our hypothesis that factor supervision transforms behavior-driven embeddings into semantically interpretable representations, enhancing both transparency and generalization.

## 4 CONCLUSION

In this work, we propose a novel LLM-driven framework for multimodal recommendation that leverages semantic factor reasoning and graph-based modeling. By extracting structured attributes from raw visual and textual content, we construct heterogeneous item-item graphs that enable interpretable and fine-grained relational reasoning. Extensive experiments demonstrate the superiority and robustness of our method across various benchmarks. In future work, we plan to explore user-side semantic graph construction, and investigate retrieval-augmented prompting with external knowledge to further enrich multimodal reasoning.

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

# A RELATED WORK

## A.1 MULTIMODAL RECOMMENDATION

Early multimodal recommenders relieved data sparsity by adding pre-training visual or textual features to ID embeddings, treating content as side information for classical collaborative filtering He & McAuley (2016); Kang et al. (2017); Chen et al. (2019). To model richer crossmodal structure, later work propagated content on interaction graphs such as MMGCN Wei et al. (2019) and GRCN Wei et al. (2020), or latent semantic graphs such as LATTICE Zhang et al. (2021), and further introduced intention-aware mechanisms such as in DualGNN Wang et al. (2021) and diffusion-based modeling in DiffMM Jiang et al. (2024), which disentangle user intents and produce noise-robust item representations. A parallel line tackles misalignment and modality noise. AlignRec Liu et al. (2024c) formulates hierarchical alignment objectives, SMORE Ong & Khong (2025) performs spectral filtering, and contrastive frameworks such as MMGCL Yi et al. (2022), SLMRec Tao et al. (2022), and BM3 Zhou et al. (2023b) build augmented views to learn modality-invariant semantics. Knowledge-transfer techniques like PromptMM Wei et al. (2024c) distill large teachers into lightweight students. Several methods also explore adaptive weighting mechanisms to dynamically adjust the influence of each modality during training. Across domains including short-form video Han et al. (2022), music Xu et al. (2023), and social platforms Yu et al. (2022), the research trajectory is shifting from simple feature fusion to noise-resilient alignment-sensitive frameworks that fully exploit heterogeneous signals while maintaining personalization Yang et al. (2023a); Yang & Yang (2024); Liang et al. (2022); Guo et al. (2022). Still, challenges remain in moving beyond surface-level fusion to semantic-level understanding, where the model could comprehend when, why, and how specific semantic factors from each modality influence user preferences.

## A.2 LLMS FOR MULTIMODAL RECOMMENDATION

Multimodal recommendation has significantly advanced with the emergence of large language models (LLMs) Ren et al. (2024); Liao et al. (2024); Wei et al. (2024b), which offer unified reasoning and generation capabilities across diverse modalities such as text, image, and video Li et al. (2023); Wang et al. (2024a); Bao et al. (2023). Traditional multimodal recommenders often rely on simple feature fusion, while recent LLM-based approaches leverage pre-trained generative knowledge to interpret, align, and personalize multimodal signals Ren et al. (2024); Wei et al. (2024b); Liao et al. (2024). Recent studies adopt various strategies for multimodal integration and adaptation. For example, MLLM4Rec Wang et al. (2024b) and Rec-GPT4V Liu et al. (2024d) prompt LLMs with visual and textual signals to align user preferences and item content, while MM-REACT Yang et al. (2023b) coordinates LLMs with expert vision modules for reasoning over multimodal inputs.

Beyond prompt tuning, the literature also explores pretraining paradigms (e.g., reconstructive, contrastive, and autoregressive) for recommendation-specific multimodal representations Yue et al. (2023); Wang et al. (2023); Zhang et al. (2023). A new direction integrates collaborative semantics with pretrained representations through domain-adaptive pretraining or quantized semantic tokenization Zhu et al. (2024). Adapter tuning and modular fine-tuning approaches further reduce computational overhead in real-world deployments Geng et al. (2023); Liu et al. (2024a). Moreover, recent advancements have enabled multimodal generation for recommendation, such as personalized news headline writing, ad copy generation, and visual storylines Liu et al. (2024b); Shen et al. (2024). Although these methods explore LLM-enhanced generation and alignment, their paradigm differs from our reasoning graph-structured approach centered on interpretable semantic factors, and thus fall outside the scope of our direct comparison.

# B METHOD

## B.1 PRELIMINARIES

We consider a set of users $\mathcal{U}$ and a set of items $\mathcal{I}$, with $|\mathcal{U}|$ users and $|\mathcal{I}|$ items, respectively. The user-item interactions are represented as a bipartite graph $\mathcal{G} = (\mathcal{V}, \mathcal{E})$, where $\mathcal{V} = \mathcal{U} \cup \mathcal{I}$ and an edge $(u, i) \in \mathcal{E}$ exists if the user $u$ has interacted with the item $i$. The interaction matrix $R \in \{0, 1\}^{|\mathcal{U}| \times |\mathcal{I}|}$ encodes this relationship, where $r_{ui} = 1$ denotes positive feedback and 0 otherwise.

To incorporate rich semantic information, each item $i \in \mathcal{I}$ is associated with a set of multimodal features $\mathbf{Z}_i = \{\mathbf{z}_i^m\}_{m \in \mathcal{M}}$, where $\mathcal{M}$ denotes the set of modalities such as vision (v), text (t) and ID (id). Each modality-specific feature $\mathbf{z}_i^m \in \mathbb{R}^{d_m}$ encodes information on the item $i$ under modality $m$, with feature dimensions $d_m$ often much larger than the embedding dimension $d$. Our objective is to model both the structural interactions and the multimodal content of items.

## B.2 CROSS-MODAL VIEW CONSISTENCY REGULARIZATION

Despite the benefits of multi-view item representations derived from ID, visual, and textual features, the inconsistent distribution across modalities often leads to representational misalignment. For example, visual representations might cluster by color or texture, while textual embeddings could follow syntactic semantics, and ID embeddings are shaped by collaborative signals. Such heterogeneity can cause instability during training and hinder the model's ability to learn robust, generalizable item embeddings. To address this, we introduce a lightweight yet effective regularization strategy: cross-view consistency loss.

### B.2.1 CROSS-MODAL VIEW CONSISTENCY LOSS

Let $\mathbf{H}_i^{\mathrm{id}}, \mathbf{H}_i^{\mathrm{v}}, \mathbf{H}_i^{\mathrm{t}}$ denote the representations of item $i$ obtained from the ID, visual, and textual graphs, respectively. Although each view contributes distinct knowledge, we expect their embeddings to maintain a consistent semantic anchor in the latent space.

To this end, we enforce pairwise alignment among views via mean squared error:

$$\mathcal{L}_{\mathrm{view}} = \frac{1}{|\mathcal{I}|} \sum_{i \in \mathcal{I}} \left[ \|\mathbf{H}_i^{\mathrm{id}} - \mathbf{H}_i^{\mathrm{v}}\|^2 + \|\mathbf{H}_i^{\mathrm{id}} - \mathbf{H}_i^{\mathrm{t}}\|^2 + \|\mathbf{H}_i^{\mathrm{v}} - \mathbf{H}_i^{\mathrm{t}}\|^2 \right]. \tag{14}$$

This loss encourages the model to learn a shared manifold across modalities, thereby enhancing representational robustness.

### B.2.2 THEORETICAL MOTIVATION AND PRACTICAL BENEFITS

From a theoretical standpoint, our consistency loss can be viewed as a form of representation alignment, minimizing the distributional discrepancy between latent spaces induced by different modalities:

$$\mathcal{L}_{\mathrm{view}} \propto \sum_{(m_1, m_2)} \mathbb{D}_{\mathrm{MSE}} \left( P(\mathbf{H}_i^{m_1}) \| P(\mathbf{H}_i^{m_2}) \right), \tag{15}$$

where $\mathbb{D}_{\mathrm{MSE}}$ is a symmetric divergence (i.e., squared L2 distance), and $m_1, m_2 \in \{\mathrm{id}, \mathrm{v}, \mathrm{t}\}$. By minimizing this divergence, we constrain the optimization landscape and implicitly guide the model toward a more coherent, multi-modal embedding space.

## B.3 SEMANTIC-ENHANCED MULTI-OBJECTIVE LEARNING

The primary goal of our model is to learn a user preference scoring function $s(f_u(u), g_i(i))$ that accurately distinguishes positive interactions from negative ones. To this end, we adopt a pairwise learning framework and minimize a main recommendation loss function $\mathcal{L}_{\mathrm{rec}}$, with the widely used Bayesian Personalized Ranking (BPR) loss.

To jointly enhance semantic alignment and representation robustness, we integrate the auxiliary semantic prediction task and the view consistency regularization into the final optimization objective, defined as:

$$\mathcal{L} = \mathcal{L}_{\mathrm{rec}} + \lambda \cdot \mathcal{L}_{\mathrm{sem}} + \eta \cdot \mathcal{L}_{\mathrm{view}}, \tag{16}$$

where $\lambda$ and $\eta$ are hyperparameters controlling the relative importance of auxiliary signals.

## B.4 THEORETICAL JUSTIFICATION OF THE SEMANTIC PREDICTION TASK

**Preliminaries.** Let $\mathbf{z}_i^{\mathrm{raw}} \in \mathbb{R}^{d_{\mathrm{raw}}}$ be the multimodal input of item $i$ and $\mathbf{y}_i \in \{0, 1\}^{|\mathcal{A}|}$ its multi-hot semantic-factor vector. The encoder produces an embedding $\mathbf{H}_i = f_\theta(\mathbf{z}_i^{\mathrm{raw}}) \in \mathbb{R}^d$ that is later used both for recommendation and for the auxiliary semantic-prediction head $\psi$.

**Information Bottleneck Objective.** Adopting the *Information Bottleneck* (IB) principle Tishby et al. (2000), the optimal representation maximises semantic relevance while suppressing nuisance information:

$$\max_{\theta,\psi} \; I(\mathbf{H}_i; \mathbf{y}_i) \; - \; \beta \, I(\mathbf{H}_i; \mathbf{z}_i^{\text{raw}}), \tag{17}$$

where $I(\,\cdot\,;\,\cdot\,)$ denotes mutual information and $\beta > 0$ balances prediction fidelity against compression.

**Variational Lower Bound on $I(\mathbf{H}; \mathbf{y})$.** Mutual information between a discrete target $\mathbf{y}$ and a continuous latent $\mathbf{H}$ can be lower-bounded through a variational distribution $q_\psi(\mathbf{y} \mid \mathbf{H})$ parameterised by the semantic head $\psi$:

$$\begin{aligned}
I(\mathbf{H}; \mathbf{y}) &= H(\mathbf{y}) - H(\mathbf{y} \mid \mathbf{H}) = H(\mathbf{y}) + \mathbb{E}_{\mathbf{H},\mathbf{y}}\big[\log q_\psi(\mathbf{y} \mid \mathbf{H})\big] \\
&\geq H(\mathbf{y}) - \mathcal{L}_{\text{sem}}(\theta, \psi),
\end{aligned} \tag{18}$$

where the *multi-label binary cross-entropy*

$$\mathcal{L}_{\text{sem}} = -\sum_{j=1}^{|\mathcal{A}|} \left[ y_{ij} \log \hat{y}_{ij} + (1 - y_{ij}) \log(1 - \hat{y}_{ij}) \right], \tag{19}$$

$$\hat{\mathbf{y}}_i = q_\psi(\mathbf{y} \mid \mathbf{H}_i)$$

acts as a *negative* lower bound on $I(\mathbf{H}; \mathbf{y})$. Minimising equation 19 therefore *maximises* the first term in equation 17.

**Implicit Compression of $I(\mathbf{H}; \mathbf{z}^{\text{raw}})$.** The encoder $f_\theta$ first aggregates information via factor-level graphs and a gated fusion module:

$$\mathbf{H}_i = \underbrace{\alpha_i^{(\text{struct})} \mathbf{H}_i^{\text{struct}}}_{\text{behavioural}} + \sum_{k=1}^{K} \underbrace{\alpha_i^{(k)} \mathbf{H}_i^{(f_k)}}_{\text{semantic}}, \tag{20}$$

$$\sum_{k=1}^{K} \alpha_i^{(k)} + \alpha_i^{(\text{struct})} = 1$$

Because each $\mathbf{H}_i^{(f_k)}$ is computed by passing messages only within items sharing the *same* factor value, modality-specific noise that does *not* conform to LLM-derived semantic structure is attenuated. Consequently, $I(\mathbf{H}_i; \mathbf{z}_i^{\text{raw}})$ is implicitly driven downward, realising the compression term in equation 17 without an explicit regulariser.

**Connection to the Training Objective.** Combining the above, the overall loss

$$\mathcal{L} \; = \; \mathcal{L}_{\text{rec}} + \lambda \, \mathcal{L}_{\text{sem}} + \eta \, \mathcal{L}_{\text{view}} \tag{21}$$

implements an empirical surrogate of equation 17:

- $\mathcal{L}_{\text{sem}}$ *maximises* $I(\mathbf{H}; \mathbf{y})$ up to a constant, grounding the representation in human-interpretable semantics;
- The graph-guided encoder design reduces $I(\mathbf{H}; \mathbf{z}^{\text{raw}})$, with $\mathcal{L}_{\text{view}}$ providing an additional regulariser that further discourages modality-specific artefacts.

Thus the auxiliary semantic task is theoretically justified as an instantiation of the Information Bottleneck, ensuring that the learned embeddings are both *discriminative* for semantic factors and *robust* to irrelevant noise.

## B.5 COMPUTATIONAL COMPLEXITY ANALYSIS

In this section, we provide a detailed analysis of the computational complexity of MARS, the overall training-time complexity of MARS can be decomposed into three main components:

$$\mathcal{O}(\text{MARS}) = \mathcal{O}(H \cdot K \cdot N \cdot F) + \mathcal{O}\left(B \cdot \sum_{k=1}^{K} V_k\right) + \mathcal{O}(B \cdot F), \tag{22}$$

where:

- $H$: Number of GNN layers
- $K$: Number of semantic graphs (factors)
- $N$: Number of items
- $F$: Hidden dimension size
- $B$: Batch size
- $V_k$: Number of attribute values in factor $k$

The first term $\mathcal{O}(H \cdot K \cdot N \cdot F)$ corresponds to the graph propagation across $K$ semantic factor graphs, where each graph involves shallow GCN operations. The second term $\mathcal{O}(B \cdot \sum_{k=1}^{K} V_k)$ represents the computational cost of the auxiliary semantic factor prediction task, and the third term $\mathcal{O}(B \cdot F)$ accounts for the cross-modal consistency loss computation.

Notably, the complexity scales *linearly* with both the number of items ($N$) and the number of semantic factors ($K$), ensuring good scalability for large-scale recommendation scenarios. The graph propagation operations are performed independently across different semantic graphs (factor-wise), making the computation efficiently parallelizable across layers without increasing depth or parameters. Moreover, MARS adopts single-step message passing on each semantic graph, followed by lightweight gated fusion, which keeps the computational overhead manageable compared to deep multi-hop graph propagation methods.

## C  EXPERIMENT

In this section, we conduct comprehensive experiments to evaluate the effectiveness and robustness of our proposed framework. Specifically, we aim to answer the following research questions: (**RQ1**) How does our method perform compared to state-of-the-art multimodal models? (**RQ2**) Can our model effectively handle cold-start scenarios while maintaining competitive quality? (**RQ3**) What is the impact of key components on overall performance? (**RQ4**) How sensitive is the model to key hyperparameters such as semantic supervision weight and embedding dimension? (**RQ5**) Does our method reduce inter-modal representation divergence and enhance consistency across semantic views? (**RQ6**) Can our framework reveal factor-aware semantic structure in the learned embedding space under explicit supervision?

### C.1  EXPERIMENTAL SETTINGS

#### C.1.1  DATASETS

We evaluate our framework on three standard subsets of the Amazon Review dataset—*Baby*, *Sports and Outdoors*, and *Clothing, Shoes and Jewelry*. Following the widely adopted 5-core preprocessing strategy, we retain users and items with at least five interactions. For the visual modality, we directly use raw product images from the original metadata. For the textual modality, we construct raw input text by concatenating available fields such as *title*, *description*, *brand*, and *category*. Rather than using handcrafted or pretrained features, we feed these raw multimodal inputs into LLMs to perform structured semantic reasoning. The statistics of datasets are summarized in Table 3.

#### C.1.2  BASELINES

We compared MARS with the following baselines. Collaborative filtering methods: **BPR** Rendle et al. (2012) and **LightGCN** He et al. (2020). Multimodal methods: **VBPR** He & McAuley (2016), **MMGCN** Wei et al. (2019), **GRCN** Wei et al. (2020), **DualGNN** Wang et al. (2021), **SLMRec** Tao et al. (2022), **LATTICE** Zhang et al. (2021), **BM3** Zhou et al. (2023b), **FREEDOM** Zhou & Shen (2023), **DiffMM** Jiang et al. (2024), **MMIL** Yang & Yang (2024), **AlignRec** Liu et al.

Table 3: Descriptive statistics of the three datasets.

| Dataset | User | Item | Interactions | Density |
|---------|------|------|--------------|---------|
| Sports | 35,598 | 18,357 | 256,308 | 0.039% |
| Clothing | 39,387 | 23,033 | 237,488 | 0.026% |
| Baby | 19,445 | 7,050 | 139,110 | 0.101% |

(2024c), **SMORE** Ong & Khong (2024). LLM-based recommendation baselines: **RecFormer** Li et al. (2023), **TALLRec** Bao et al. (2023), **A-LLMRec** Kim et al. (2024) and **UniMP** Wei et al. (2024a).

### C.1.3 EVALUATION METRICS

We evaluate top-$K$ recommendation performance using Recall@$K$ and NDCG@$K$, which measure hit rate and ranking quality, respectively. Datasets are split into training, validation, and testing sets with an 8:1:1 ratio under the 5-core protocol. We follow the all-ranking evaluation setting, ranking each user against all candidate items. The model with the best Recall@20 on the validation set is selected for final testing.

### C.1.4 IMPLEMENTATION DETAILS

We implement our framework using PyTorch and build upon the MMRec Zhou (2023) code-base to ensure consistent and reproducible comparisons with prior multimodal recommendation methods. For all baseline models, we adopt the optimal hyperparameter configurations as reported in their original implementations or papers. For our proposed method, we tune the loss weights for the semantic supervision ($\lambda$), and cross-modal consistency ($\eta$) from the candidate set $\{0.1, 0.01, 0.001, 0.0001\}$. For semantic factor reasoning, we employ the GPT-4o model as the backbone LLM to extract structured semantics from raw inputs. We finally adopt the Top-10 most frequent semantic factors from GPT-4o's output for processing textual and visual inputs. All experiments are conducted on a single NVIDIA A40 GPU (48GB).

## D DECLARATION ON THE USE OF LARGE LANGUAGE MODELS

In the preparation of this work, the authors used GPT-4o and other GPT-series models for two specific purposes. First, GPT-4o was employed within our methodological framework to extract semantic factors from text and image inputs, as illustrated in Fig. 1. Second, GPT models were used to polish the writing and improve grammatical correctness. After using these tools, the authors reviewed and edited the content as needed and take full responsibility for the content of the publication.

