# OpenReview forum: "Cross-Modal Factor Reasoning with LLMs: Toward Semantic-Structured Generalization for Recommendation"
_ICLR.cc/2026/Conference — Submitted to ICLR 2026_

### Official Review · Reviewer_MvG2 · 2025-10-18

**Soundness:** 2
**Presentation:** 2
**Contribution:** 2
**Rating:** 4
**Confidence:** 5

**Summary:**

This paper proposes MARS, a novel multimodal recommendation framework that leverages Large Language Models (LLMs) for cross-modal factor reasoning. The key idea is to extract human-interpretable semantic factors (e.g., functionality, material) from raw multimodal content using LLMs, construct heterogeneous graphs based on these factors, and integrate semantics into representation learning via an auxiliary semantic prediction task and cross-modal consistency loss. Extensive experiments on Amazon datasets demonstrate superior performance over state-of-the-art baselines in both accuracy and cold-start scenarios.

**Strengths:**

- Effectively integrates LLM-derived factors into graph-based learning.

- Relevant experiments for the cold start scenario are provided.

**Weaknesses:**

- [Mandatory] How are the item complementarity, stylistic coherence, or functional substitutability mentioned in the Introduction reflected? In fact, the selected Amazon datasets (Baby, Sports, Clothing) do not exhibit such phenomena. Please carefully review the meta-data of these datasets.

- [Mandatory] In the Introduction, the authors claim that no work considers leveraging collaborative signals to support content. In reality, some works [1-2] have used collaborative signals to purify content. Other works [3-4] have also considered using collaborative signals to guide modality alignment.

- [Mandatory] The factor extraction in Eq. 1 requires different adjustments for different scenarios. The current experiments focus on the e-commerce domain but lack experiments in other domains to verify generalizability. Additionally, its quality heavily depends on the pre-defined factors in the prompt.

- [Mandatory] While the factor extraction strategy captures key features to some extent, it undoubtedly loses much fine-grained information about items, reducing the personalization of the recommendation system.

- [Mandatory] This design significantly hinders the extension to multiple modalities. For example, in scenarios involving audio modalities (e.g., TikTok datasets), it may fail to extract effective factors from spectrograms.

- [Mandatory] Admittedly, current multimodal recommendation systems cannot fully utilize visual information. However, converting visual modalities into text modalities undoubtedly results in the loss of valuable semantic information, limiting the theoretical upper bound of multimodal recommendation performance.

- [Mandatory] This model is heavily dependent on the capabilities of LLMs.

- [Mandatory] The configuration of $\mathcal{A}_{f_k}$ determines the quality of the heterogeneous semantic graph construction and introduces higher hyperparameter tuning costs compared to semantic homogeneous graphs.

- [Mandatory] The training of the Semantic Factor Prediction Auxiliary Task is supposed to focus on the semantic encoder. However, it fails to retain semantic-relevant information while filtering out modality-specific noise.

- [Mandatory] Some works published in 2024–2025 should be included as baselines, such as [2–5].

- [Mandatory] Splitting into multiple factors increases the amount of graph message passing, so the model's efficiency needs to be empirically and theoretically validated and discussed.



Refs:

[1] Multi-View Graph Convolutional Network for Multimedia Recommendation, ACM MM 2023.

[2] Cohesion: Composite graph convolutional network with dual-stage fusion for multimodal recommendation, SIGIR 2025.

[3] Mentor: multi-level self-supervised learning for multimodal recommendation, AAAI 2025.

[4] GUME: Graphs and User Modalities Enhancement for Long-Tail Multimodal Recommendation, CIKM 2024.

[5] Mind Individual Information! Principal Graph Learning for Multimedia Recommendation, AAAI 2025.

**Questions:**

Please refer to Weaknesses. Btw, I have some optional questions:

- [Optional] Can this work handles modalities like audio without losing critical information?

---

### Official Review · Reviewer_tVZn · 2025-10-27

**Soundness:** 3
**Presentation:** 2
**Contribution:** 2
**Rating:** 4
**Confidence:** 4

**Summary:**

This paper proposes MARS, a framework for cross-modal factor reasoning with LLMs, to solve existing multimodal recommendation limitations (shallow fusion, no structured semantics, disconnected collaborative-content signals) . It uses LLMs to extract interpretable semantic factors (e.g., functionality) from visual/textual content, builds heterogeneous graphs for item semantic relations, and adds an auxiliary semantic prediction task plus cross-modal consistency loss .

**Strengths:**

1. Unlike traditional multimodal recommendation approaches that rely on low-level similarity (e.g., visual/textual embedding proximity) and lack high-level relations (e.g., functionality, style coherence), MARS uses LLMs (e.g., GPT-4o) to extract interpretable semantic factors from raw visual/textual content and builds heterogeneous graphs based on these factors, enabling fine-grained semantic relation modeling .

2. MARS introduces an auxiliary semantic prediction task to align collaborative embeddings with LLM-derived factors, and a cross-modal consistency loss to unify ID/visual/textual representations. This resolves the one-way supervision gap (collaborative signals only for ranking) in existing frameworks, making embeddings semantically grounded .

**Weaknesses:**

1. MARS relies on powerful LLMs (e.g., GPT-4o) for semantic factor extraction, which may incur high computational costs and depend on external LLM services. No lightweight alternatives for factor reasoning are proposed, limiting deployment in resource-constrained scenarios.

2. Experiments are only conducted on three Amazon Review subsets (Sports, Clothing, Baby), all belonging to e-commerce. No validation on other multimodal recommendation domains (e.g., social media micro-videos, music) means its adaptability to diverse scenarios remains unproven.

3. Some of the text in Figure 1 is too small and difficult to read.

**Questions:**

1. Does the method require powerful LLMs (e.g., GPT‑4o) during inference?

---

### Official Review · Reviewer_TA2u · 2025-11-01

**Soundness:** 2
**Presentation:** 4
**Contribution:** 3
**Rating:** 4
**Confidence:** 4

**Summary:**

This paper proposes MARS, a semantic-structured multimodal recommendation framework that aims to move beyond shallow feature fusion. The authors use LLMs to extract human-interpretable semantic factors (e.g., functionality, material, scene) from item content, construct a factor-aware heterogeneous graph to model structured semantic relations across items, and leverage collaborative filtering signals to supervise semantic factor learning. This enables the model to align content semantics with preference semantics. Experiments on multiple Amazon datasets show improved performance, cold-start gains, and enhanced interpretability through more meaningful semantic clustering.

**Strengths:**

1. The paper highlights an important gap in multimodal recommendation: content semantics (what an item is) does not necessarily align with preference semantics (why users choose it). Positioning collaborative filtering as a semantic value filter to ground LLM-extracted semantics in actual user preference is insightful and intellectually valuable.

2. Using LLMs to extract explicit semantic factors and building a factor-aware heterogeneous graph brings human-readable structure into item representation learning. This improves transparency and provides more intuitive explanations than typical embedding-based multimodal fusion.

3. The proposed factor-aware semantic graph offers a more principled way to capture high-level item relations that go beyond surface-level visual/textual similarity. This design aligns with how users mentally categorize items, enabling more meaningful reasoning compared to pure embedding-similarity graphs.

**Weaknesses:**

1. The paper assumes multimodal encoders mainly learn surface-level similarity (e.g., appearance), but recent VLMs such as CLIP/BLIP-2/LLaVA already capture functional and conceptual semantics. The authors should clarify what semantic gaps remain unsolved by modern multimodal encoders and why factor-graph modeling is necessary beyond them.

2. While results show improvements, there is no head-to-head comparison against strong VLM-based baselines (e.g., CLIP-enhanced or BLIP-enhanced recommenders). Without such comparisons, it is difficult to isolate whether gains come from the factor graph itself or from general architectural complexity.

3. The ablation (w/o FP) shows that semantic factor prediction benefits ranking performance, which indirectly supports the usefulness of CF-supervised semantics. However, the paper does not provide direct evidence that collaborative filtering supervision refines the semantic relevance of the extracted factors or filters out non-preference-relevant semantics. The claim that CF acts as a “semantic value filter” remains conceptual. Additional analyses—such as semantic factor quality evaluation, preference relevance studies, visualization of factor distributions before/after CF supervision, or semantic pruning impact—would strengthen this key contribution.

**Questions:**

1. Could authors specify which aspects of semantics current VLMs fail to capture that your factor-graph modeling resolves? A comparison against a CLIP/BLIP-augmented baseline would make the necessity of your approach more convincing.

2. Can authors provide evidence that CF supervision improves semantic relevance of extracted factors rather than merely boosting ranking performance?

---

### Official Review · Reviewer_HYsP · 2025-11-03

**Soundness:** 2
**Presentation:** 3
**Contribution:** 2
**Rating:** 2
**Confidence:** 3

**Summary:**

This paper proposes the MARS, which extracts interpretable semantic factors from multimodal content through LLM, builds heterogeneous graphs, and introduces semantic prediction and cross-modal consistency tasks, which improve the performance of recommendation systems in cold start and generalization scenarios. This method sounds good, but lacks codes.

**Strengths:**

+ The paper was well written and clearly structured.
+ Sufficient comparisons on many types of baseline models.
+ The design of the semantic prediction task is interesting.

**Weaknesses:**

+ GPT-4o is used to extract semantic factors, the adaptability of different LLMs (such as open source models, small language models) is not evaluated, and the impact of LLM reasoning costs on large-scale scenario deployment is not discussed.
+ The selection of semantic factors relies on the Top-10 high-frequency factors output by LLM. There is a lack of quantitative analysis on the rationality of factor screening, and the impact of different factor combinations on model performance has not been verified.
+ The experimental dataset is limited to subsets of Amazon, and the scenario is quite single.
+ Source code is missing.

**Questions:**

+ The current method relies on the quality of semantic factors output by LLM, does not consider possible deviations or errors in LLM, and lacks an automatic evaluation or error correction mechanism for the quality of factors.
+ What is the motivation of the current factor selection method? Are there cases where some low-frequency semantic factors that are critical to a specific user group/item category are missed?
+ For different scenarios, the definition of semantic factors may need to be adjusted. How to extend it to other fields, such as video recommendation?
+ The same factor may be defined differently in different types of goods. Will this affect the accuracy of semantic relationship modeling in heterogeneous graphs? Such as the functionality factor “outdoor” in clothing and tools.

---

### Meta-Review · Area_Chair_kPBx · 2025-12-03

**Summary:**

This paper proposes the MARS framework, which uses LLMs to extract semantic factors from item content, builds factor-aware heterogeneous graphs, and introduces semantic prediction and cross-modal consistency objectives to improve multimodal recommendation.

**Strengths**

* The paper is clearly written and well organized.
* Incorporating human-readable semantic factors allows better interpretability.
* The attempt to bridge content semantics and preference semantics is conceptually interesting.

**Weakness**

* Strong dependency on GPT-4-level LLMs is not addressed, with no evaluation of smaller/open-source alternatives and unclear inference-time cost for deployment.
* Generalization is insufficiently demonstrated: all experiments use Amazon subsets, and the method lacks validation in other domains including video, audio, social content, etc.
* Claims regarding novelty of using collaborative filtering to refine content semantics are factually incorrect; prior work already addresses this.
* Semantic factor extraction is under-validated, relying on high-frequency factor sampling rather than principled selection or factor quality evaluation.
* No comparisons against modern VLM-powered recommenders (CLIP/BLIP-based), leaving open whether performance gains arise from factor reasoning or simply model capacity.
* Mandatory methodological objections from Reviewer MvG2, regarding dataset appropriateness, factor granularity, modality scalability, encoder supervision quality, and efficiency, were not addressed and remain unresolved due to absence of rebuttal.

Collectively, these issues undermine the paper’s empirical validity and contributions, and particularly the absence of rebuttal to mandatory concerns makes it unsuitable for acceptance at this time.

**Reviewer Concerns:**

No rebuttals are provided.

**Reviewer Scores:**

All reviewers will keep scores, since no rebuttals are provided.

---

### Decision · Program_Chairs · 2026-01-26

Reject